# Functional Analysis of the Major Pilin Proteins of Type IV Pili in *Streptococcus sanguinis* CGMH010

**DOI:** 10.3390/ijms25105402

**Published:** 2024-05-15

**Authors:** Yi-Ywan M. Chen, Yuan-Chen Yang, Hui-Ru Shieh, Yu-Juan Lin, Wan-Ju Ke, Cheng-Hsun Chiu

**Affiliations:** 1Department of Microbiology and Immunology, College of Medicine, Chang Gung University, Taoyuan 333, Taiwan; cowcow119@hotmail.com (H.-R.S.); ndhuonedirection@gmail.com (Y.-J.L.); ruru650316@gmail.com (W.-J.K.); 2Graduate Institute of Biomedical Sciences, College of Medicine, Chang Gung University, Taoyuan 333, Taiwan; funnyamylike@gmail.com; 3Molecular Infectious Disease Research Center, Chang Gung Memorial Hospital, Linkou 333, Taiwan; chchiu@cgmh.org.tw

**Keywords:** *Streptococcus sanguinis*, type IV pilus, major pilins, twitching motility, biofilm, invasion

## Abstract

The *pil* gene cluster for Type IV pilus (Tfp) biosynthesis is commonly present and highly conserved in *Streptococcus sanguinis*. Nevertheless, Tfp-mediated twitching motility is less common among strains, and the factors determining twitching activity are not fully understood. Here, we analyzed the functions of three major pilin proteins (PilA1, PilA2, and PilA3) in the assembly and activity of Tfp in motile *S. sanguinis* CGMH010. Using various recombinant *pilA* deletion strains, we found that Tfp composed of different PilA proteins varied morphologically and functionally. Among the three PilA proteins, PilA1 was most critical in the assembly of twitching-active Tfp, and recombinant strains expressing motility generated more structured biofilms under constant shearing forces compared to the non-motile recombinant strains. Although PilA1 and PilA3 shared 94% identity, PilA3 could not compensate for the loss of PilA1, suggesting that the nature of PilA proteins plays an essential role in twitching activity. The single deletion of individual *pilA* genes had little effect on the invasion of host endothelia by *S. sanguinis* CGMH010. In contrast, the deletion of all three *pilA* genes or *pilT*, encoding the retraction ATPase, abolished Tfp-mediated invasion. Tfp- and PilT-dependent invasion were also detected in the non-motile *S. sanguinis* SK36, and thus, the retraction of Tfp, but not active twitching, was found to be essential for invasion.

## 1. Introduction

Type IV pili (Tfp), a structural homolog of the type II secretion system [1], are stretchable filaments present on the surface of archaea and bacteria [2]. Tfp are subdivided into Types IVa and IVb based on the leader sequence of the prepilin and the size of the mature major pilin [3]. Both types of Tfp have been extensively studied in Gram-negative bacteria [4]. Type IVa Tfp were recently identified in Gram-positive bacteria, including *Clostridium* spp. and *Streptococcus sanguinis* [5], though Type IVb Tfp are found only in Gram-negative bacteria. Genes encoding the proteins required for Type IVa Tfp biosynthesis are typically scattered in a few clusters in Gram-negative bacteria [5] and *Clostridium* spp. [6,7], while in *S. sanguinis*, the *pil* genes are arranged as an operon [8,9].

Tfp are made of major and minor pilins; the major pilins are present in thousands of copies in a pilus, whereas the minor pilins are present in lower abundance [10]. Some minor pilin proteins constitute an initiation complex to prime Tfp biosynthesis, and others act as ligands for interaction with host cells [11,12,13]. Notably, the Gram-negative systems generally feature a single major pilin protein, whereas *S. sanguinis* commonly harbors two to three homologous major pilin proteins [14]. Additionally, the Tfp system contains two ATPases for extension (PilB) and retraction (PilT/U) of the Tfp, respectively. The continuous action of PilB and PilT/U activates surface-dependent twitching motility [15]. Note that the nomenclature of the *pil* genes is not standardized; therefore, in this manuscript, we follow the convention used for *Pseudomonas aeruginosa* Tfp genes.

Tfp are known to mediate surface-dependent twitching motility, genetic transformation, surface sensing, adherence, and biofilm formation [4,16]. The functional significance of Tfp has been extensively studied in Gram-negative pathogens, including *Neisseria meningitidis* [17] and *P. aeruginosa* [18,19], while in Gram-positive systems, where Type IVa Tfp have not been widely explored, similar functions have also been described [8,9,14,20,21]. Gram-negative bacteria employ a single system for all observed activities, whereas Gram-positive bacteria utilize two independent Tfp systems for twitching motility and natural competence. Specifically, *Streptococcus pneumoniae* utilizes Tfp comprising Com proteins for DNA uptake during genetic transformation but not for surface-dependent twitching motility, despite the Tfp being morphologically identical to the motility-driven Tfp [22]. On the other hand, *S. sanguinis* utilizes a Com system for DNA uptake [23] and the Tfp produced by the *pil* operon for twitching motility and the adherence of host cells [8,9,14]. Similarly, the Tfp of *Clostridium* spp. enable twitching motility and biofilm formation [6,7,24,25,26], but are unable to take up extracellular DNA. Finally, the study by Martini et al. demonstrates that *S. sanguinis* SK36 can invade endothelial cells via the activity of Tfp, highlighting the crucial role of Tfp in pathogenesis [21], as commonly observed in Gram-negative pathogens.

*S. sanguinis* is a dominant oral isolate and an early colonizer of dental plaque [27]. Advances in genomic studies have confirmed that the *pil* operon is commonly present in the genome of *S. sanguinis* strains. Interestingly, *S. sanguinis* is the only streptococcal species harboring the *pil* operon, suggesting that Tfp could provide specific advantages for *S. sanguinis*. Although the arrangement and sequence of the *pil* genes are conserved, the number of major pilin genes (*pilA*) varies across strains, with two or three *pilA* genes being observed. For example, among the most-studied *S. sanguinis* strains, strains SK36 and CGMH010 contain three major pilin genes, *pilA1*, *pilA2*, and *pilA3*, whereas strain 2908 contains two major pilin genes: *pilE1* and *pilE2*. Most of the *pil* genes share a sequence identity of approximately 90% or higher at the deduced amino acid level among homologs across strains, though loci encoding the major and minor pilins exhibit less conservation [14]. Previously, we examined the sequences of the *pilA* genes of 30 clinical isolates and found that although *pilA* genes are relatively less conserved between strains, significant identities between PilA proteins are present within a strain. Specifically, strains harboring three *pilA* genes typically display over 90% identity between two of the three *pilA* paralogs, while the identity with the remaining paralog ranges from 65% to 85%, suggesting that gene duplication occurred during evolution. Furthermore, a study with *S. sanguinis* 2908 demonstrated that a single major pilin can generate motile Tfp, but Tfp composed of PilE1 exhibit a faster velocity than Tfp made of PilE2 [20]. Whether all PilA proteins in the three-*pilA* system are required for the biosynthesis of functional Tfp and whether the Tfp-mediated activities are affected by the PilA proteins remain unexplored.

Our recent functional studies on two *S. sanguinis* strains encoding three *pilA* paralogs, the motile CGMH010 and non-motile SK36, confirmed that Tfp are essential for optimal adherence to host cells, regardless of motility [14]. Here, we investigated the role of the three highly homologous major pilins of CGMH010 and found that each protein exhibits different effects on the structure and activity of Tfp.

## 2. Results

### 2.1. Sequence Analysis of the PilA Proteins and Generation of Anti-PilA Antisera

PilA1 and PilA3 of *S. sanguinis* CGMH010 share 94% identity at the deduced amino acid level, and PilA2 shares 72% and 70% identity with PilA1 and PilA3, respectively. In agreement with the structural studies of the major pilin proteins [28], the N-terminal regions of PilA1, PilA2, and PilA3 are highly conserved and form an α-helix (α1). Specifically, PilA1 and PilA3 are identical from amino acids 1 to 53, whereas four conserved and two semi-conserved substitutions are present in PilA2. The helix-breaking amino acid, Pro-22, is also present in all three PilA proteins (Figure 1A). The C-terminal region, containing sequences that form the structurally conserved global domain, is less conserved between the PilA proteins. The predicted 3-D structures obtained from the AlphaFold protein structure database also indicate that all PilA proteins exhibit a characteristic “lollipop” architecture [3], with the global domain embedded in the C-terminal half of the α1 (Figure 1B).

To generate antibodies that are specific to each of the PilA proteins, a peptide unique to PilA1 was used to generate a PilA1-specific antiserum (Figure 1A, red box). This antiserum does not cross-react with PilA2 and PilA3. However, the antiserum generated using a histidine-tagged PilA2 protein (His-PilA2) cross-reacts with both PilA1 and PilA3 (with reduced intensity), whereas the antiserum generated by using a histidine-tagged PilA3 protein (His-PilA3) cross-reacts with PilA1 (Figure 1C). Although the anti-PilA2 and anti-PilA3 antisera exhibit cross-reactivity, it is sufficient to detect specific PilA proteins when analyzing the results obtained from all three antisera.

### 2.2. The Effect of PilA Proteins in the Assembly of Tfp in S. sanguinis CGMH010

To analyze the function of each of the PilA proteins in the structure and the function of Tfp, recombinant *S. sanguinis* strains with single, double, or triple deletions of the *pilA* genes were generated in *S. sanguinis* CGMH010 using non-polar *erm* or *kan*, as detailed in the Materials and Methods (Figure 2A). Western blot analysis with anti-PilA antisera demonstrated the presence of non-cognate PilA proteins in total cell lysates of all *pilA*-deficient derivatives, confirming that mutations were non-polar (Appendix A).

To analyze whether all three PilA proteins are required for the assembly of surface filaments, extracellular Tfp were purified from various strains and examined by transmission electron microscopy (TEM). Filaments were readily detected in the preparations of wild-type CGMH010, strains Δ*pilA2*, Δ*pilA3*, Δ*pilA23*, and Δ*pilT* (Figure 2B). Filaments were also detected in strain Δ*pilA1*, but the filaments were more difficult to find, suggesting that the loss of *pilA1* affected filament formation. Additionally, the filaments derived from the Δ*pilA2* and Δ*pilA3* strains were similar to those of wild-type CGMH010, and the filaments purified from the Δ*pilA1* and Δ*pilA23* strains were shorter than those of the wild-type CGMH010. No filaments were found in Δ*pilA13* and Δ*pilA123*, and only short, aggregated filaments were detected in the preparation from the Δ*pilA12* strain. Strain Δ*pilT*, deficient in the retraction ATPase, generated long filaments. To confirm that the observed filaments were made of PilA proteins, the preparations were also subjected to immuno-gold labeling with anti-PilA2 or anti-PilA3 antiserum (Figure 2B). All observed filaments, regardless of the length, were the products of PilA proteins. Thus, PilA1 and PilA3 alone could generate extracellular filaments, but pili made of PilA3 (strain Δ*pilA12*) differed from pili derived from PilA1 (strain Δ*pilA23*), despite the high degree of homology between PilA1 and PilA3.

To further confirm the composition of the extracellular pili in the *pilA-*deletion mutants, the extracellular pilus preparations were subjected to Western blot analysis with PilA-specific antisera (Figure 3). Anti-CodY antiserum [29] was utilized as a control to ensure that the preparations were free of cytoplasmic proteins. Extracellular pilin proteins were detected in all strains except Δ*pilA13* and Δ*pilA123*, indicating that the deletion of both *pilA1* and *pilA3* would abolish, or greatly reduce, Tfp biosynthesis.

### 2.3. The Twitching Activity of Tfp Composed of Different PilA Proteins

Next, we wanted to examine the impact of PilA proteins on Tfp-mediated twitching motility on an agar surface by using the above-described *pilA-*deletion derivatives of CGMH010. Our results demonstrated that the deletion of *pilA1* resulted in the most significant reduction in the twitching zone on TH agar, compared to wild-type CGMH010 (Figure 4A). Among the strains without *pilA1* (Δ*pilA1*, Δ*pilA12*, Δ*pilA13*, and Δ*pilA123*), only the Δ*pilA1* strain expressed a marginal twitching zone, whereas no twitching zone was observed in the Δ*pilA12* and Δ*pilA13* strains. On the other hand, the deletion of *pilA2* and *pilA3* had little effect on the size of the twitching zone, indicating that PilA2 and PilA3 were dispensable for motile Tfp biosynthesis. The critical role of PilA1 in generating twitching-active pili was further confirmed in strain Δ*pilA23*, which generated a twitching zone smaller than that of the wild-type strain, but larger than that of strain Δ*pilA1*. As expected, the pilin-null strain (Δ*pilA123*) failed to generate a visible twitching zone. Taken together, these results suggest that PilA1 alone can generate a twitching-positive Tfp, whereas PilA2 and PilA3 together cannot fully compensate for the loss of *pilA1* in the assembly of twitching-active Tfp.

As the growth rate of all PilA mutant strains was comparable to that of the wild-type CGMH010, presumably, strains that can move at high velocity would generate larger twitching zones compared to strains moving at low velocity after prolonged incubation. To verify this hypothesis, we analyzed the velocity of a single chain of cells by microscopic examination. Using a wet-mount preparation [14], it was found that the bacterial chains of a strain did not move at the same speed (Figure 4B); presumably, chain length would affect the sensitivity of detection. Nevertheless, in agreement with macroscopic examination, the median velocity of the motile strains (Δ*pilA2* and Δ*pilA3*) was more similar to the wild-type CGMH010 than to the non-motile strains (Δ*pilA12*, Δ*pilA13*, Δ*pilA123*, and Δ*pilT*). Strain Δ*pilA23*, which generated a reduced twitching zone on the agar surface, exhibited a wide range of velocity, with a higher number of slow-moving chains compared to strains Δ*pilA2* and Δ*pilA3*, resulting in a lower median velocity. Although strain Δ*pilA1* generated a marginal twitching zone, most chains of Δ*pilA1* expressed limited motility, confirming that PilA1 was critical for the biosynthesis of motile Tfp.

### 2.4. Wild-Type Tfp Were Essential for Maintaining Biofilm Structure

Previously, we found that the inactivation of *pilT* (the ∆*pilT* strain), which abolishes twitching activity, reduced biofilm formation in a flow-cell system where constant shear forces act upon the biofilm [14]. The decreased biofilm formation in ∆*pilT* may be due to interference from longer Tfp, compared to wild-type CGMH010, produced in the absence of PilT (Figure 2B). To further analyze the impact of twitching activity on biofilm formation, we examined the ability of the *pilA* mutant strains (described above) to form biofilms in a flow-cell system. Confocal laser scanning microscope (CLSM) examination of areas close to the medium intake port revealed that the thickness and structure of biofilms generated by the motile strains, i.e., ∆*pilA2*, ∆*pilA3*, and ∆*pilA23*, were comparable to those of wild-type CGMH010. In contrast, the biofilm derived from strains without active twitching motility, i.e., ∆*pilA1*, ∆*pilA12*, ∆*pilA13*, and ∆*pilA123*, was thinner compared to biofilms generated by wild-type CGMH010 (Figure 5). Thus, Tfp-mediated twitching motility is required for optimal biofilm formation under constant shearing forces.

### 2.5. Motile Tfp Are Essential for the Invasion of Host Cells by S. sanguinis

A recent study by Martini et al. indicated that Tfp mediate the invasion of endothelial cells by *S. sanguinis* SK36, a non-motile strain, suggesting that invasion is independent of twitching motility [21]. To investigate whether Tfp-mediated invasion is common to all *S. sanguinis* strains, and to identify the role of twitching motility in this process, we analyzed the invasion of human coronary artery endothelial (HCAEC) cells by *S. sanguinis* CGMH010 and its *pilA-* and *pilT*-deletion derivatives. *S. sanguinis* CGMH010 and SK36 invaded HCAEC cells at a similar rate in a PilA protein-dependent manner (Figure 6). Strains lacking *pilT* were drastically reduced in their ability to invade HCAEC cells. The ∆*pilA1*, ∆*pilA2*, and ∆*pilA3* strains of CGMH010 displayed comparable levels of invasion, while conversely, the Tfp-null strain (∆*pilA123*) could not invade HCAEC, confirming that filament structure plays a role in invasion. Thus, the ability to retract is essential for Tfp-mediated invasion, regardless of whether a twitching zone could be detected on an agar surface.

## 3. Discussion

The study in *S. sanguinis* 2908 indicated that motile pili can be generated by a single major pilin protein, suggesting a redundancy of major pilin proteins in Tfp assembly; However, the degree of Tfp-mediated twitching motility is dependent upon the major pilin proteins that compose Tfp [20]. In contrast, our results demonstrate that the three PilA proteins expressed by *S. sanguinis* CGMH010 exhibit diverse impacts on the assembly and function of Tfp. Specifically, we show that PilA1 is most critical for Tfp production, and, although PilA1 and PilA3 share a high level of amino acid sequence identity, neither PilA3 alone nor a combination of PilA2 and PilA3 could fully compensate for the loss of *pilA1*. These results indicate that PilA1 and PilA3 function differently in Tfp biosynthesis. In addition, the results of this study support the hypothesis that Tfp-mediated motility is required for optimal biofilm formation under constant shearing force, and that PilT activity is essential for the Tfp-mediated invasion of host cells, regardless of whether strains could generate a twitching zone on an agar surface.

Recent reconstruction studies using cryo-electron microscopy on the major pilins of *N. meningitidis*, *N. gonorrhoeae*, and *P. aeruginosa* indicated that, rather than being continuous, the α-helix contains a melted region between two conserved helix-breaking residues, Gly14 and Pro22 [30,31]. It is proposed that the melting of this region could facilitate the integration of the pilin subunit into the growing Tfp, and a fully extended conformation of this region could allow the pilus to reversibly elongate up to three times its original length [30]. Thus, this region is likely to play a critical role in the assembly, extension, and retraction of Tfp. A comparison of all major pilin proteins from both *S. sanguinis* CGMH010 and *S. sanguinis* 2908 revealed that these major pilins share an identical sequence at the α1 region, except for the PilA2 of CGMH010. Although all major pilin proteins harbor Pro22, but not Gly14, PilA2 contains three conserved substitutions flanking Pro22. It is tempting to suggest that the altered amino acids may reduce the activity of PilA2 in the assembly of filaments, and, in fact, PilA2 had a minor effect on the assembly of retractable filaments.

Studies have shown that the primary sequence of the C-terminal region of the pilin protein, which contains sequences of an αβ-loop, a 4-stranded antiparallel β-sheet, and a variable D-loop, is less conserved among strains and species [4,28]. Using I-TASSER (https://zhanggroup.org/I-TASSER/, accessed on 16 February 2024), 4-, 2-, and 3-stranded β-sheets were predicted in PilA1, PilA2, and PilA3, respectively, suggesting that subtle differences in the structures could exist between these proteins. As the *pil* genes are arranged in an operon, and all three *pilA* genes contain a putative ribosomal binding site at an appropriate position, the different activities between PilA proteins in the assembly of Tfp should be related to the sequence, but not the expression level of the proteins. Similar to variations in Tfp-mediated twitching motility between *S. sanguinis* strains, a recent study in *Acidovorax citrulli* found that PilA proteins of strains belonging to different groups differ in sequence, and Tfp made of different PilA proteins exhibit distinct activities in twitching, biofilm formation, and even interspecies competitive abilities [32]. Furthermore, similar to the Tfp system of *S. sanguinis*, *Haloferax volcanii* produces multiple major pilins for Tfp biosynthesis, and Tfp composed of different major pilins exhibit different functions in the process of biofilm formation [33]. Thus, although PilA proteins are highly conserved in a *S. sanguinis* strain, the subtle sequence differences in the PilA proteins contribute significantly to the activity of the Tfp.

Unlike Gram-negative pathogens, in which Tfp-mediated twitching motility plays a critical role in virulence [4], past and current studies have revealed that Tfp, but not twitching motility, are more critical for *S. sanguinis* in its primary niche, the oral cavity. This notion is supported by the following observations: first, only a portion of the *S. sanguinis* isolates express twitching motility on an agar surface [14,34]; second, Tfp-mediated adherence [8,14] and invasion of host cells (this study) are observed in both motile and non-motile *S. sanguinis* strains. Interestingly, our recent sequence study on 30 clinical *S. sanguinis* strains from the Chang Gung Memorial Hospital (CGMH) bacteria bank found only one strain carrying two major pilin genes, indicating that strains encoding three major pilin genes are more prevalent. Like *S. sanguinis* 2908, the CGMH strain harboring two major pilin genes also expresses twitching motility. As the nature of the major pilins and the composition of Tfp could determine twitching activity, it is tempting to suggest that the major pilin gene underwent duplication during evolution, resulting in the production of three major pilin genes. The presence of the third PilA protein may lead to the production of Tfp expressing different levels of twitching motility. Since Tfp-mediated motility may not provide significant advantages for *S. sanguinis* to survive in the oral cavity, perhaps the non-motile Tfp evolved for energy conservation. On the other hand, a recent study in *P. aeruginosa* revealed that Tfp-mediated twitching motility promotes the surface departure of *P. aeruginosa* under reduced shearing forces. In contrast, high shearing forces enhance the adhesion of bacterial cells by counteracting Tfp retraction [35]. This observation raises the possibility that twitching motility serves different roles for *S. sanguinis* in the environment under high shearing forces, i.e., the bloodstream, and low shearing forces, i.e., the oral cavity. Our previous study on 81 clinical *S. sanguinis* CGMH strains revealed a total of 44 strains that could express twitching activity on either TH agar or blood agar, representing 54% of the isolates [14], which is higher than the incidence of the oral isolates reported by Henriksen et al. (49%) [34]. Thus, one could speculate that Tfp-mediated motility could provide additional benefits for *S. sanguinis* to establish outside the oral cavity.

Initially, we tried to determine the differences in the Tfp morphology between the *pilA* mutant strains by examining surface filaments on whole cells by TEM. However, *S. sanguinis* generates multiple surface hair-like structures, which is common in oral streptococci [36], and thus, we were unable to distinguish the differences in Tfp between *S. sanguinis* strains with confidence. On the other hand, although the *pilT-*deletion strain is routinely used as the non-motile control, the reduced invasion by Δ*pilT* may be caused by the lengthy Tfp, but not the motility per se. Nevertheless, loss of *pilT* abolished invasion by both motile (CGMH010) and non-motile (SK36) strains, and the ability to retract, but not the production of a twitching zone, is critical for Tfp-mediated invasion by *S. sanguinis*. Similarly, as non-motile *pilA-*knockout strains of CGMH010 did not express long Tfp, the reduced biofilm formation of these strains was unrelated to the length of the Tfp.

The significance of expressing multiple highly homologous major pilin proteins in *S. sanguinis* is yet undefined, although sequence variations and post-translational modification of the pilin proteins may contribute to antigenic variation and immune evasion [30,37,38]. However, the antigenic variation from the highly identical amino acid sequences of the PilA isomers, especially PilA1 and PilA3, remains to be elucidated. The results from this study suggest that Tfp provide competitive advantages for *S. sanguinis*, similar to their role in Gram-negative pathogens, and further demonstrate that the nature of the major pilin proteins plays an essential role in the twitching activity of *S. sanguinis* CGMH010.

## 4. Materials and Methods

### 4.1. Bacterial Strains and Culture Conditions

The bacterial strains and plasmids used in this study are listed in Table 1. *S. sanguinis* strains were routinely cultivated in Todd Hewitt (TH) medium at 37 °C in a 10% CO_2_ atmosphere or in an anaerobic jar. Where indicated, erythromycin (Em) at 10 μg mL^−1^ or kanamycin (Km) at 500 μg mL^−1^ was included for the selection of recombinant *S. sanguinis* strains. Recombinant *E. coli* strains were cultivated in LB broth containing Km at 50 µg mL^−1^, with continuous shaking at 37 °C.

### 4.2. Generation of pilA Knockout Derivatives of S. sanguinis CGMH010

All mutant derivatives of *S. sanguinis* CGMH010 were generated by ligation mutagenesis [40]. The primers used in this study are listed in Appendix A. To generate a specific knockout, two fragments, 5′ and 3′ to the target gene, respectively, were generated by PCR with specific primers. Restriction endonuclease recognition sequences were included in the primers to facilitate subsequent ligation reactions. The PCR products were digested and mixed with DNA fragments containing the non-polar Em resistance gene (*erm*) [41] or non-polar Km resistance gene (*kan*) [42] in a reaction that allows for the ligation of the 5′ flanking fragment, followed by the *erm* or *kan* fragment and then the 3′ flanking fragment. For the generation of strains Δ*pilA1*, Δ*pilA2*, Δ*pilA3*, Δ*pilA12*, Δ*pilA23*, and Δ*pilA123*, the ligation mixture was used to transform *S. sanguinis* CGMH010 [43] with selection for Em resistance. The Δ*pilA13* strain was generated by transforming a ligation mixture containing the *pilA3* inactivation construct into the Δ*pilA1* strain and selecting for Km resistance. The allelic exchange event in the isogenic knockout strains was confirmed by colony PCR with primers located outside the *erm* or *kan* insertion site with specific primers.

### 4.3. Expression of the Pilin Proteins and Preparation of Pilin Protein-Specific Antibody

An anti-PilA1 antibody was generated in rabbits immunized with an internal peptide of PilA1 containing amino acid residues 95 to 104 (AKGGDKIADA) (Figure 1, red box). The peptide and an antiserum specific to the peptide were generated commercially (GenScript, Piscataway, NJ, USA).

A histidine-tagged PilA2 protein (His-PilA2) was constructed using the pET28a(+) expression plasmid (Novagen, Darmstadt, Germany), and purified His-PilA2 was used for anti-PilA2 antiserum production in rabbits. Briefly, the coding sequence for amino acids 38 to 152 of PilA2 was amplified from *S. sanguinis* CGMH010 by PCR (primer sequences in Appendix A), cloned into pET28a(+), and established in *E. coli* BL21. The sequence of the recombinant plasmid was verified by sequencing. Induction by IPTG and the purification of His-PilA2 by Ni^2+^ affinity chromatography were performed under denaturing conditions using the standard procedure (Qiagen, Hilden, Germany). The identity of the purified His-PilA2 protein was confirmed by Matrix-Assisted Laser Desorption/Ionization–Time of Flight Mass Spectrometry. Approximately 2.5 mg of purified His-PilA2 protein was separated on 12% SDS-PAGE gel, and the expected protein band was excised from the gel and used to generate polyclonal antiserum in the rabbits (LTK BioLaboratories, Taoyuan, Taiwan).

A similar approach was used to generate a histidine-tagged PilA3 (His-PilA3). The coding sequence for amino acid residues 38 to 148 of PilA3 (primer sequences in Appendix A) was cloned in plasmid pET28a(+), expressed, purified, and confirmed as described above, and then used for antiserum production in rabbits.

To examine the specificity and titer of the anti-PilA1 antiserum, a histidine-tagged PilA1 protein (amino acid residues 38 to 147; His-PilA1) was also prepared using the same approach.

### 4.4. Protein Structure Prediction

The 3D structures of the PilA proteins were predicted by using the AlphaFold protein structure database (https://alphafold.ebi.ac.uk, accessed on 29 April 2024).

### 4.5. Purification of Extracellular Tfp

Extracellular Tfp of *S. sanguinis* strains were purified as previously described with minor modifications [20]. Briefly, cultures (100 mL) of *S. sanguinis* strains were grown in TH medium to an O.D._600_ of 0.8, harvested, washed once with 10 mM NaPO_4_, and then concentrated 100-fold in pilus buffer (20 mM Tris-HCl [pH 7.6], 50 mM NaCl). The concentrated suspensions were vortexed at full speed for 2 min to shear Tfp. At the end of the shearing step, the suspensions were centrifuged at 6000× *g* for 10 min and the supernatant was recovered. The sheared filaments in the supernatant were recovered by ultra-centrifugation at 100,000× *g*, 4 °C, for 1 h. The final pellet was dissolved in 60 μL pilus buffer. A total of 15 μL of the final suspension was separated on 12% SDS-PAGE, and 5 μL of the preparation was used for TEM examination.

### 4.6. Immunogold Labeling, Negative Staining, and TEM Observation

For transmission electron microscopy (TEM) examination, 5 μL of the extracellular Tfp preparation was applied onto a 200-mesh carbon-coated copper grid (Agar, Essex, UK) and allowed to set for 30 s. The sample was subjected to negative staining directly or immunogold labeling with anti-PilA antibody, followed by negative staining. Negative staining of Tfp preparations was performed by adding 5 μL 2% methylamine tungstate (Nanoprobes, New York, NY, USA) to the sample, allowing it to set for 10 s, and then placing the sample in a moisture-proof box (44% relative humidity) for 24 h before TEM examination. The grid was observed using a JEM-2100 Plus electron microscope (JEOL, Japan, Tokyo).

For immunogold labeling, the sample was fixed on the grid with 10 μL of a fixing solution composed of 0.2% glutaraldehyde and 4% paraformaldehyde at room temperature for 5 min. At the end of the fixing reaction, the grid was washed three times with pilus buffer to remove the fixing solution. The sample was then blocked with TBST (150 mM NaCl, 50 mM Tris [pH 7.6], 0.5% Tween 20) containing 3% BSA at room temperature for 30 min, followed by incubation with anti-PilA2 or anti-PilA3 antiserum at 1:10 dilution in TBST for 1 h. The unbound antibody was removed by rinsing with drops of the pilus buffer. The bound antibody was recognized by goat anti-rabbit IgG conjugate with 10 nm gold particles (Sigma, St. Louis, MI, USA) at room temperature for 1 h. The grid was again rinsed with pilus buffer 10 times to remove the secondary antibody prior to negative staining and TEM examination.

### 4.7. Gel Electrophoresis and Western Blot Analysis

Protein preparations were separated on 12% SDS-PAGE for Western blot analysis. The separated proteins were transferred from gels to PVDF membranes (Merck, Darmstadt, Germany). After blocking with 5% skim milk in TBST at 4 °C overnight, the blot was hybridized with antibodies specific to each of the pilin proteins in TBST containing 5% skim milk for 1 h at room temperature. The anti-PilA1 antibody was used at 1:1000; the anti-PilA2 and anti-PilA3 antisera were used at 1:5000. Horseradish peroxidase-conjugated anti-rabbit antibody (GeneTex, Irvine, CA, USA) and luminol-based Immobilon western chemiluminescent HRP substrate (Millipore, Darmstadt, Germany) were used to detect bound PilA-specific antisera. The results were imaged using a UVP BioSpectrum Imaging system (Vilber, Marne-la-Vallée, France).

### 4.8. Examination of Twitching Motility

Twitching motility was examined macroscopically on TH agar (1%) as previously described [14]. Briefly, *S. sanguinis* strains were inoculated in straight lines on TH agar (1%) and incubated in an anaerobic jar with 30 mL water added to the bottom of the jar (to maintain humidity) at 37 °C for 66 h.

Wet-mount preparations were used for microscopic examination of twitching motility, as previously described, with minor modifications [14]. An 8 μL aliquot of log-phase cultures (O.D._600_ = 0.4) of *S. sanguinis* strains grown in TH medium was used to prepare the wet-mount. The wet-mount sample was incubated at 37 °C in a 10% CO_2_ atmosphere for 30 min before microscopic examination at 100× magnification (Olympus BX41, Tokyo, Japan). The movement of the bacteria was recorded for 10 s using a TrueChrome AF microscope camera (Tucsen, Fuzhou, China). The velocity of bacterial movement was analyzed using ImageJ (https://imagej.net/) with a particle-tracking plugin (MultiTracker). The velocity distribution is presented as a violin plot created using GraphPad Prism 7.

### 4.9. Preparation of Flow-Cell Biofilm and Examination by CLSM

Biofilms in the flow-cell system were prepared as previously described [14]. Briefly, overnight cultures of *S. sanguinis* strains grown in biofilm medium (BM) [44] containing 40 mM glucose (BMG) were diluted in BMG to an O.D._600_ of 0.4. A total of 300 μL of the diluted culture was injected into the growth chamber. The chamber was kept upside down without medium flow for 4 h at 37 °C. The medium flow was set to 5 mL^−1^ h^−1^. The chamber was incubated for 5 h at 37 °C. At the end of the incubation, the biofilms were stained with SYTO 9 and propidium iodide (PI) for 30 min using a LIVE/DEAD biofilm viability kit (Invitrogen, Carlsbad, CA, USA). After staining, the samples were washed with 500 μL double-distilled H_2_O. The stained biofilms were examined by confocal laser scanning microscopy (CLSM) with a 63× oil immersion objective lens (LSM780, Zeiss, Jena, Germany). The images were displayed by the ZEN acquisition software (Zen 2012 SP5 FP3 black, Zeiss).

### 4.10. Invasion Assay

Primary human coronary artery endothelial cells (HCAEC, ATCC, Manassas, VA, USA) were grown in vascular cell basal medium (ATCC) supplemented with an Endothelial Cell Growth Kit—VEGF (ATCC). HCAEC cells were seeded in 24-well plates and grown for 12 h before the assay. Cells were infected with exponential-phase cultures (O.D._600_ = 0.4) of *S. sanguinis* strains at an MOI of 20 for 2 h. At the end of incubation, the extracellular bacteria were removed by washing with PBS twice, followed by a penicillin (100 units mL^−1^)–gentamicin (300 μg mL^−1^)–streptomycin (100 μg mL^−1^) treatment at 37 °C for 1 h. At the end of the treatment, the cells were washed with PBS twice, then lysed with 0.1% Triton X-100 for 10 min to recover the intracellular bacteria. Intracellular bacterial counts were determined by serial dilution and plating. The invasion rate was calculated as (CFU recovered/CFU of the inoculum) × 100%. Significant differences between strains were analyzed using one-way ANOVA followed by Tukey’s test. Differences were considered significant if *p* < 0.01.

## 5. Conclusions

Multiple major pilin proteins are commonly observed in *S. sanguinis* Tfp. Although the major pilin proteins share high levels of identity within a strain, presumably as a result of gene duplication, these pilin proteins contribute differently to the assembly of motile Tfp. As the Tfp-mediated adherence and invasion of host cells are observed in both motile and non-motile *S. sanguinis* strains, Tfp-mediated twitching motility may provide limited advantages for *S. sanguinis* competitiveness. Thus, the production of various major pilin proteins may allow *S. sanguinis* to modulate twitching motility for energy conservation.

## Figures and Tables

**Figure 1 ijms-25-05402-f001:**
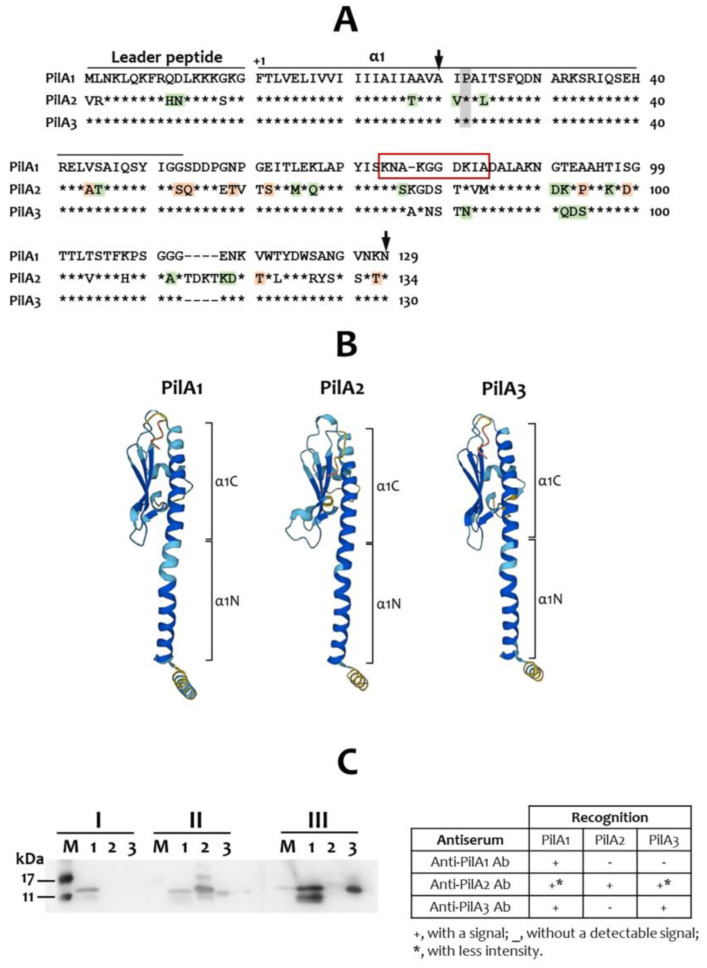
PilA proteins of *S. sanguinis* CGMH010 and antibody production. (**A**) Alignment of the deduced amino acid sequence of PilA1, PilA2, and PilA3 is shown. Sequences of the leader peptide and the α1 region are overlined. Stars indicate identical amino acids compared to PilA1 at the same position. Conservative and semi-conservative substitutions are shaded in green and orange, respectively. The conserved Pro22 is shaded in grey. The region selected for producing polyclonal PilA1-specific antiserum is boxed in red. The regions for generating polyclonal PilA2- and PilA3-specific antisera are indicated by vertical arrows. (**B**) The 3D structure prediction by AlphaFold. The predicted structures contain the leader peptides. The model confidence scores (pLDDT) are indicated by colors. Cobalt blue, pLDDT > 90; sky blue, pLDDT > 70; yellow, pLDDT > 50; orange, pLDDT < 50. (**C**) One microgram of purified recombinant His-PilA proteins was separated on a 12% SDS-PAGE gel and hybridized with anti-PilA1 (I), anti-PilA2 (II), and anti-PilA3 (III) antisera. Lanes 1, 2, and 3 are recombinant His-PilA1, His-PilA2, and His-PilA3 proteins, respectively. M, a protein marker, was loaded. The specificity of each antiserum is summarized in the right-hand table.

**Figure 2 ijms-25-05402-f002:**
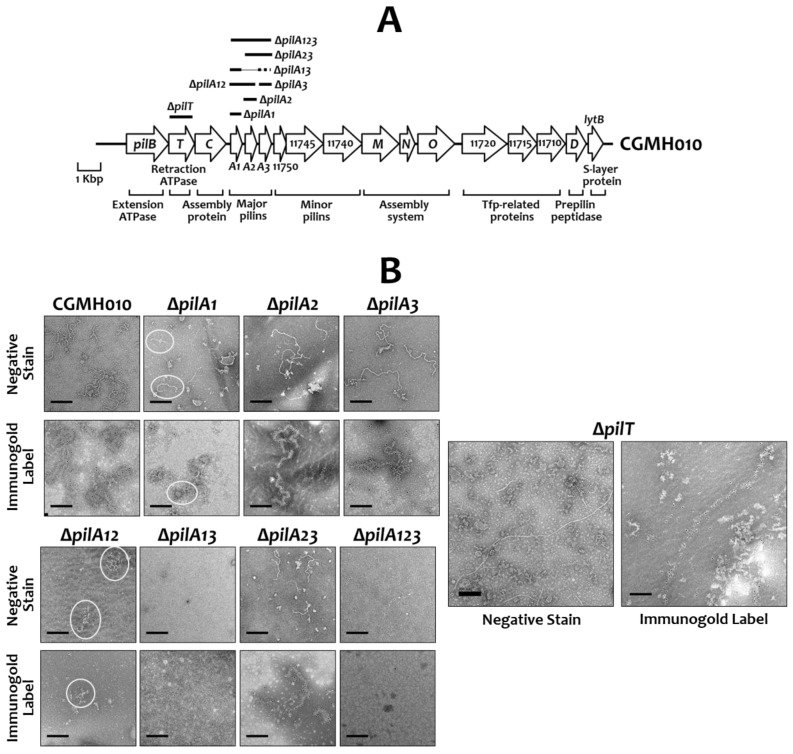
Tfp composed of isomeric PilA proteins differ in morphology. (**A**) Illustration of the genotypes of the deletion derivatives of CGMH010. The *pil* cluster and the gene names or FFV08_tag numbers are listed within or below the gene. The regions replaced by non-polar *erm* and *kan* in the mutant strains are indicated by a thick solid and dashed horizontal line, respectively, and listed above the genes. (**B**) TEM examination of the extracellular pilus preparations of *S. sanguinis* strains. The preparations were negatively stained or immunogold-labeled with anti-PilA3 (wild-type CGMH010 and strains Δ*pilA1*, Δ*pilA2*, Δ*pilA12*, Δ*pilA23*, and Δ*pilT*) or anti-PilA2 (strains Δ*pilA3*, Δ*pilA13*, and Δ*pilA123*) antisera, followed by negative staining. Samples were observed by TEM at 25,000 or 30,000× magnification. The short filaments purified from Δ*pilA1* and Δ*pilA12* are circled. The black line is the scale bar (200 nm).

**Figure 3 ijms-25-05402-f003:**
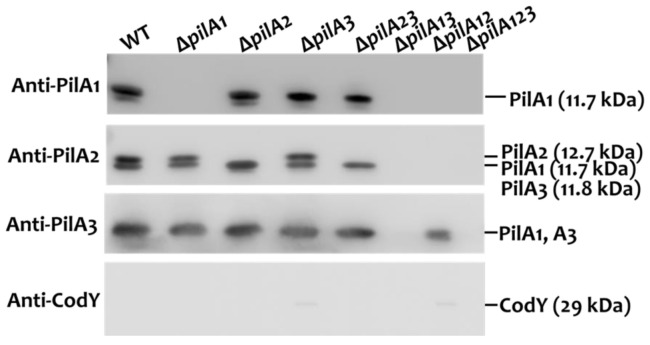
Western blot analysis of the extracellular pilin proteins of *S. sanguinis* CGMH010 and its *pilA*-deletion derivatives. An equal volume of the extracellular pili preparation from each strain was separated on 12% SDS-PAGE and analyzed by immunoblotting. The anti-serum used in each blot is shown to the left of the blot. The positions of all proteins and their molecular weight are shown on the right.

**Figure 4 ijms-25-05402-f004:**
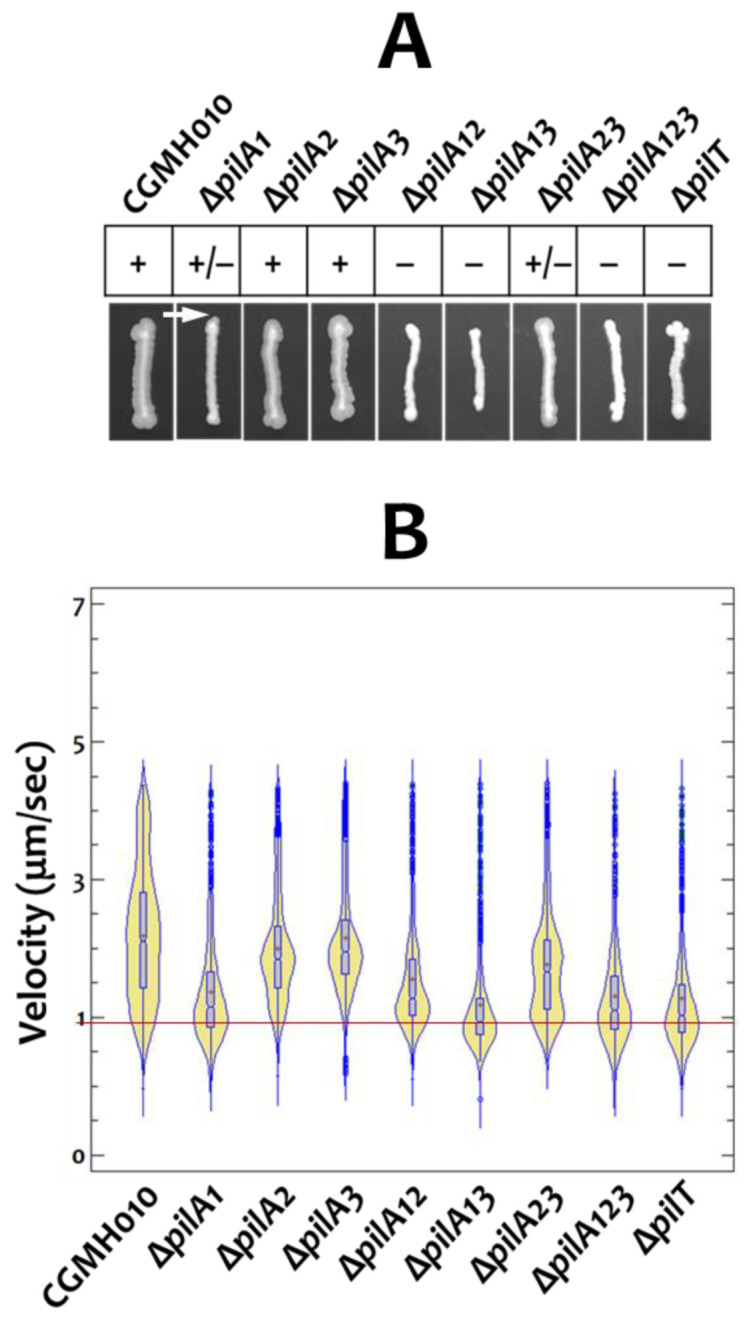
Macroscopic and microscopic examination of the twitching motility of *S. sanguinis* CGMH010 and its *pilA*-deletion derivatives. (**A**) Macroscopic examination of twitching activity. *S. sanguinis* CGMH010 and its *pilA-*deficient derivatives were inoculated on TH agar and incubated for 66 h. +, expression of a twitching zone; −, no detectable twitching zone; +/−, expression of a reduced twitching zone (compared to wild-type CGMH010). The spreading area generated by strain Δ*pilA1* is indicated by an arrow. (**B**) Microscopic examination of twitching activity. The motility of *S. sanguinis* strains was examined microscopically, and the velocity distribution of approximately 1000 chains is demonstrated by a violin plot. Average velocity is indicated by a red + sign, and the median is indicated by a horizontal bar in each sample. The basal level of detection, derived from strain Δ*pilT*, is indicated by a red horizontal line.

**Figure 5 ijms-25-05402-f005:**
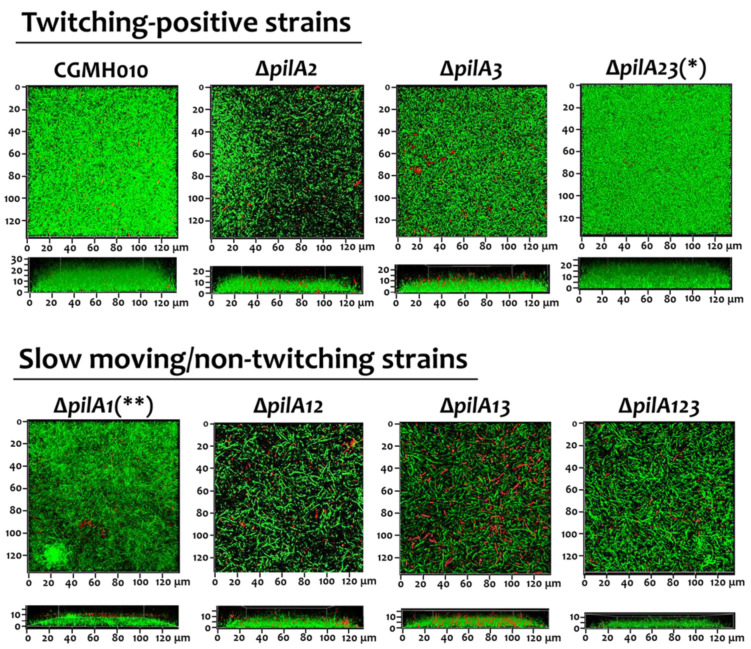
CLSM examination of biofilm cultures of *S. sanguinis* CGMH010 and its *pilA*-deletion derivatives in a flow-cell system. Biofilms underwent LIVE/DEAD staining (Invitrogen) and were examined at 64× magnification. Top and side views are shown. Green fluorescence from SYTO-9 staining indicates live cells, and the red fluorescent stain (PI) indicates dead cells. *, indicates that the twitching zone of the ∆*pilA23* strain on an agar surface is narrower than the wild-type CGMH010. **, the ∆*pilA1* strain produced a marginal twitching zone on the agar surface.

**Figure 6 ijms-25-05402-f006:**
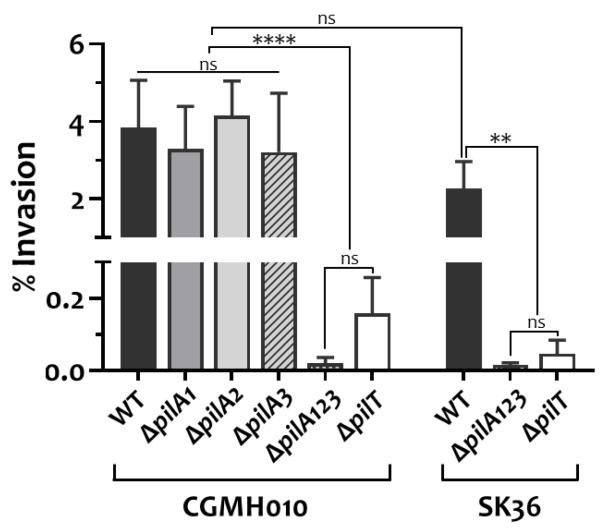
The impact of Tfp on the invasion of *S. sanguinis* strains. Invasion efficiency was calculated as (CFU adhered/CFU of the inoculum) × 100%. The numbers are the mean and standard deviation of four experiments. Significant differences between strains were analyzed using one-way ANOVA followed by a Tukey test. ****, *p* < 0.0001; **, *p* < 0.01; ns, not significant.

**Table 1 ijms-25-05402-t001:** Bacterial strains and plasmids used in this study.

Strains	Relevant Phenotype ^a^	Description	Source or Reference
*S. sanguinis* strains			
CGMH010		Clinical isolate	[14]
Δ*pilA1*	Em^R^, PilA1^−^	*pilA1* is replaced with *erm* in CGMH010	This study
Δ*pilA2*	Em^R^, PilA2^−^	*pilA2* is replaced with *erm* in CGMH010	This study
Δ*pilA3*	Em^R^, PilA3^−^	*pilA3* is replaced with *erm* in CGMH010	This study
Δ*pilA12*	Em^R^, PilA1^−^,PilA2^−^	*pilA1* and *pilA2* are replaced with *erm* in CGMH010	This study
Δ*pilA13*	Em^R^, Km^R^, PilA1^−^, PilA3^−^	*pilA1* and *pilA3* are replaced with *erm* and *kan*, respectively, in CGMH010	This study
Δ*pilA23*	Em^R^, PilA2^−^, PilA3^−^	*pilA2* and *pilA3* are replaced with *erm* in CGMH010	This study
Δ*pilA123*	Em^R^, PilA1^−^, PilA2^−^, PilA3^−^	All three *pilA* genes are replaced with *erm* in CGMH010	This study
Δ*pilT*	Em^R^, PilT^−^	Also known as CHW01, *pilT* is replaced with *erm* in CGMH010	[14]
SK36		oral isolate	[39]
Plasmids			
pET28a(+)	Km^R^	Expression vector for N-terminal His-tagged proteins	Novagen
pET28a/*pilA1*	Km^R^	pET28a(+) harboring the coding sequence of *pilA1*	This study
pET28a/*pilA2*	Km^R^	pET28a(+) harboring the coding sequence of *pilA2*	This study
pET28a/*pilA3*	Km^R^	pET28a(+) harboring the coding sequence of *pilA3*	This study

^a^, R, resistance; ^−^, deficiency.

## Data Availability

Data contained within the article.

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
