# Peer review of "Functional Analysis of the Major Pilin Proteins of Type IV Pili in Streptococcus sanguinis CGMH010"

_ijms, 2024, doi:10.3390/ijms25105402_

Round 1
Reviewer 1 Report
Comments and Suggestions for Authors
This manuscript has examined functions of pilin proteins (A1, A2 and A3) in type IV pili in Streptococcus sanguinis CGMH010. Authors have utilized deletion strains, antibodies, biofilms and functional assays delineating functional consequences. Experiments are appeared to be conducted carefully and conclusions drawn are supported by the data. I have a few minor comments that need to be addressed.
1. Line 93 and elsewhere in the article change aa to amino acids
2. Lines 98-99: “The helix-breaking aa, Ala-14 and Pro-22, are also present 98 in all three PilA proteins (Fig. 1A).” Ala is not a helix breaking amino acid. It has high propensity to form helix in proteins. Authors should amend this statement.
3. Antibody studies Fig 1B can be summarized in Table, currently it is confusing or difficult to follow which antibody cross-react with what proteins.
4. For more clarity authors should provide 3-D structures of the proteins (from pdb) in a new figure.
5. Regions used to generate polyclonal antibodies (Figure 1A) should be clear stated in terms of amino acids (residues XX to XX).
Author Response
- Line 93 and elsewhere in the article change aa to amino acids
Response:
Revised as suggested. We highlighted all changes in the revised document.
- Lines 98-99: “The helix-breaking aa, Ala-14 and Pro-22, are also present 98 in all three PilA proteins (Fig. 1A).” Ala is not a helix breaking amino acid. It has high propensity to form helix in proteins. Authors should amend this statement.
Response:
Indeed, the pilin proteins of Streptococcus sanguinis only harbor the helix-breaking amino acid Pro22, but not Gly14, different from Pseudomonas aeruginosa and Neisseria gonorrhoeae which harbor both residues. We have corrected the description in the result section (ln 98) by removing Ala14 and modified the discussion regarding the sequence variations between the pilin proteins (ln 273, and lns 277-282). Accordingly, we also removed the labeling on Ala14 in Fig. 1.
- Antibody studies Fig 1B can be summarized in Table, currently it is confusing or difficult to follow which antibody cross-react with what proteins.
Response:
As suggested, we included a table describing the specificity of each antibody in Fig. 1C (a modification of the original Fig. 1B) to enhance the readability of the revised document. To be noted, we have included the 3-D structures of the PilA proteins in Fig. 1B in the revised document, and thus, the figure demonstrating the specificity of anti-PilA antibodies is now Fig. 1C.
- For more clarity authors should provide 3-D structures of the proteins (from pdb) in a new figure.
Response:
As suggested, we have presented the 3-D structures of all three PilA proteins in Fig. 1B and revised the text accordingly (lns 99-103). The method for the 3-D structure prediction is described in the materials and methods under “Protein structure prediction” (lns 394-396).
- Regions used to generate polyclonal antibodies (Figure 1A) should be clear stated in terms of amino acids (residues XX to XX).
Response:
The region in each PilA protein for the construction of a His-tagged recombinant protein is indicated in the materials and method section, under “Expression of the pilin proteins and preparation of pilin protein-specific antibody” (lns 371-393).
Reviewer 2 Report
Comments and Suggestions for Authors
The article entitled “Functional analysis of the major pilin proteins of the Type IV pili in Streptococcus sanguinis CGMH010” tackles a very important topic, the virulence factors in Streptococcus sanguinis. Since it is an understudied microorganism, the information regarding the virulence is of the utmost importance. In the era of new technologic tools for the bacterial diagnosis in the microbiology laboratory, identification of this pathogen is on the rise. However, little it is known about the pathogenic potential a therefore this paper aims to do exactly that.
The experiments were conducted in an impeccable manner. They are explained in detail and the information provided proves to be crucial in understanding the mechanism of motility involving the pili in Streptococcus sanguinis.
In addition, the tables and figures are presented and explained in a concise manner, adding substantial value to this paper.
Their conclusion is pertinent and supported by the results they provided.
In conclusions, I am comfortable accepting this paper for publication.
It was an honour to evaluate it.
Author Response
We are pleased to hear that our study has been approved.